

# Development of a revised IHA method for the cumulative impacts of
# cascade reservoirs on flow regime
Xingyu Zhou [1, 2], Xiaorong Huang [1, 2], Hongbin Zhao [1, 2], Kai Ma [1, 2]
[1] State Key Laboratory of Hydraulics and Mountain River Engineering, Sichuan Chengdu 610065
[2] College of Water Resource & Hydropower, Sichuan University, Chengdu, Sichuan 610065
*Correspondence to*: Xiaorong Huang (hxiaorong@scu.edu.cn)
**Abstract.** The impacts of reservoirs, especially multiple reservoirs, on the flow regimes and ecosystems of rivers have received
increasing attention. The most widely used metrics to quantify the characteristics of flow regime alterations are the indicators
of hydrologic alteration (IHA) which include 33 parameters. Due to the difference in the degree of alteration and the
intercorrelation among IHA parameters, the comprehensive evaluation method that assigns the same weight to each indicator
is obviously inadequate. A revised IHA method is proposed by utilizing the projection pursuit (PP) and real-coded accelerated
genetic algorithm (RAGA). Data reliability is analysed by using the length of record (LOR) method. The projection values
reflecting the comprehensive characteristics of the evaluation parameters are calculated. Based on these methods, a scientific
and reliable evaluation of the cumulative impacts of cascading reservoirs on the flow regime was made by examining the
Jinsha River. The results showed that with the continuous construction of reservoirs, the alteration degrees of IHA parameters
gradually increased in Groups 1, 2, 3, and 4, but decreased in Group 5, and the flow duration curves showed characteristics of
a head drop and tail lift. The flow regime alteration of the outlet section was more stable than before. This change had a
negative impact on downstream fish reproduction and ecological protection. An attempt at ecological regulation was made to
simulate the natural rising process of water.
## 1 Introduction
Free-flowing rivers (FFRs) support diverse, complex and dynamic ecosystems globally, providing important societal and
economic services (Grill et al., 2019). Humans have extensively altered river systems through impoundments and diversions
to meet their water, energy, and transportation needs (Nilsson et al., 2005). Only 37% of rivers longer than 1,000 km remain
free-flowing over their entire length and 23% flow uninterrupted into the ocean. Very long FFRs are largely restricted to remote
regions of the Arctic and of the Amazon and Congo basins. From 1978(when China's reform and opening up began) to 2017,
China was experienced an unprecedented boom in the construction of dams. Until date 2017, 98795 reservoirs and dams were
built in China with a total storage capacity of 9.035×1011m3, accounting for 32% of the annual runoff of all rivers and streams
in China, of which 732 reservoirs are large reservoirs with a total capacity of 7.21×1011m3, accounting for 79.8% of the total
capcity. (2017 Statistic Bulletin on China Water Activities, 2017). Flow regulation and fragmentation of large global river


systems have received increasing attention (Best, 2019; Schmitt et al., 2018; Chen and Olden, 2017; Winemiller et al., 2016;
Nilsson et al., 2005). Flow variability is widely recognized as a primary driver of biotic and abiotic conditions in riverine
ecosystems (Poff and Zimmerman, 2010; Poff et al., 1997). However, fully understanding the cumulative impacts of multiple
dams on flow regime remains a challenge in both the scientific and management communities.

34         To evaluate the characteristics and ecological effects of flow regime changes, indicators are often needed to quantify the

extent of hydrological alterations caused by reservoirs or dams. Olden and Poff (2003) found more than 170 hydrological
indicators that can describe the different components of the flow regime and capture the ecologically relevant streamflow
attributes. However, large numbers of hydrologic metrics are too complicated to use, and many metrics are intercorrelated,
resulting in statistical redundancy (Gao, et al.,2009; Poff, and Zimmerman., 2010). Studies have sought to explore redundancy
among hydrological indicators. For example, Olden and Poff (2003), Yang et al. (2008), Gao et al. (2009) and Fantin‑Cruz
et al. (2015) used principal component analysis (PCA) to evaluate the patterns of statistical variation for each parameter and
identified a small subset of hydrological indicators as the most representative of the ecological flow regimes. Yang et al. (2017)
used the criteria importance through intercriteria correlation (CRTTIC) algorithm to remove repetition and identify the weights
of indicators. The weight of each hydrological indicator is assumed to be proportional to the standard deviation and inversely
proportional to its correlation with other indicators. Then, high-weight indicators and some low-weight indicators that have
important effects on aquatic ecology are used as representative indicators. Obviously, this selection is subjective and arbitrary.
The most widely used metrics for characterizing river flow regime changes are the indicators of hydrologic alteration (IHA),
which were developed based on 33 hydrological parameters in five groups, namely, the magnitude of monthly streamflow, the
magnitude and duration of annual extreme flows, the timing of annual extreme flows, the frequency and duration of high and
low pulses, and the rate and frequency of flow changes (Richter et al.1996; Mathews and Richter,2007). Richter et al. (1997)
proposed the range of variability (RVA) method for evaluating the degree of alteration of the hydrological flow regime with
IHA metrics. Nevertheless, intercorrelation still exists among the 33 parameters (Olden and Poff, 2003; Gao et al., 2012).
Vogel et al. (2007) proposed a small set of representative indicators, i.e., the nondimensional metrics of ecodeficit and
ecosurplus, which are based on flow duration curves (FDCs) and are computed over any time period of interest (month, season,
or year). Ecodeficit and ecosurplus reflect the overall loss or gain of streamflow resulting from flow regulation. Some studies
(Zhang et al., 2015; Gao et al., 2012; Gao et al., 2009) have demonstrated that the ecodeficit and ecosurplus metrics provide a
simplified and adequate representation of hydrological impacts, compared with the use of the more complex IHA and RVA
hydrological approaches.

58         Scholars have become increasingly concerned with the cumulative effects of multiple dams deriving from individual

dams on hydrological processes (Huang et al., 2018; Wang et al., 2018; Wen et al., 2018; Wang et al., 2017a; Deitch et al.,
2013; Santucci et al., 2005). The combined effect of cascade reservoirs on hydrological processes is cumulative and greater



than that associated with individual reservoirs (Dos Santos et al., 2018; Huang et al., 2010). In comparison, Santucci et al.
(2005) found little evidence on the cumulative effects of low head dams (< 15 in height). There are few existing studies that
compare the effects of single versus multiple dams on the hydrologic regime. (Zhao et al., 2012). Are the effects of multiple
dams additive, multiplicative, or largely insignificant? These fundamental problems always cause disturbances to scientific
and management communities (Timpe and Kaplan, 2017).

66       In summary, previous studies on method improvement were based on the statistical reduction in the dimensionality of

multi-index data and evaluated the hydrological alterations of rivers. The disadvantage is that retaining most of the information,
also leads to the loss of some information. For example, a PCA usually only maintains 80% of the data information. In this
study, a very different idea was employed. Data mining and optimization methods were used to identify the characteristics of
the indicator system and identify the difference weight of each indicator to overcome the deficiency in the comprehensive
evaluation given the same weight for each indicator. At the same time, global optimization also reduced the deviation in the
evaluation caused by intercorrelation among indicators.

73       Based on previous studies, the objectives of the present study are as follows: 1)to develop an updated    weight

determination method for IHA indicators and precisely evaluate hydrological alteration; 2) to analyze cumulative effects on
the flow regime with the construction of cascade reservoirs; and 3)to provide beneficial insights into ecological reservoir
operation and sustainable water resource management under future scenarios.
**2 Study area and data**
**2.1 Study area**

79       The Jinsha River comprises the upper reaches of the Yangtze River in China and originates from the northern foot of the

Tanggula Mountains in the Tibetan Plateau. The Jinsha River flows along a distance of approximately 3500 km and has a
drainage area of 502,000 km$^2$, which is approximately 27.8% of the entire basin area of the Yangtze River. The mean annual
precipitation in the Jinsha River basin is 710 mm and the average annual runoff is 4471 m$^3$/s. The largest tributary of the Jinsha
River is the Yalong River, and its inflow accounts for 1/3 of the total discharge of the Jinsha River. As the largest hydropower
base in China, the Jinsha River contains a hydropower resource of 112.4 million KW. At present, 20 reservoirs with
hydropower stations have been planned for development along its mainstream with 72.04 million KW of installation capacity,
and 21 reservoirs with hydropower stations have been planned for development along the Yalong River with 28.56 million
KW of installation capacity (Fig. 1). Among the 13 reservoirs built, most have a weak regulation capacity while the Ertan,
Xiangjiaba and Xiluodu reservoirs have large capacities, strong regulation abilities and great effects on the flow regime of the
outlet section of the basin (Table 1). There is a national nature reserve in China that protects rare fish from 1.8 km downstream





of the Xiangjiaba reservoir to the main stream of the Yangtze River in Masangxi, Chongqing municipality (355 km in length).
This region is also known as the national nature reserve for rare fish in the upper reaches of the Yangtze River, and its main
protection targets include three rare fish, i.e., paddlefish (Psephurus gladius), Yangtze sturgeon fish (Acipenser dabryanus),
and Chinese sucker fish (Myxocyprinus asiaticus), and 67 unique fish. This region is also an important habitat for four major
Chinese carps, i.e., black carp (Mylopharyngodon piceus), grass carp (Ctenopharyngodon idellus), silver carp
(Hypophthalmichthys molitrix), and big head carp (Aristichthysmobilis). With the successive construction of the Jinsha River
cascade reservoirs, flow regime changes will have an impact on downstream fish habitats.
**2.2 Data**
The daily streamflow data of the Panzhihua, Huatan, Pingshan and Xiangjiaba hydrological gauges were collected (Table 2).
The climate data used in this study are daily precipitation data at 28 stations from 1966-2017, and the daily inflow and outflow
data of the Xiangjiaba Reservoir and Xiluodu Reservoir were collected to analyze the effects of ecological regulation. The
Pingshan station is located approximately 28 km upstream of the Xiangjiaba Reservoir and covers 99.96% of the controlled
drainage area of the reservoir, but it has been out of service since 2012 due to construction of the Xiangjiaba Reservoir.
Therefore, data from Xiangjiaba station, which was newly built in 2012 and is located close to the Xiangjiaba Reservoir, were
supplemented with Pingshan station data(Huang et al., 2018).
**3 Method**
**3.1 Projection pursuit method**
Projection pursuit (Friedman and Tukey, 1974, 1981; Wang et al., 2017b; Wang et al., 2019) uses data mining and optimization
methods to project high-dimensional data into low-dimensional space and analyze the characteristics of high-dimensional data
through the distribution structure of low-dimensional projection data. The main steps are as follows:
(1) We normalize all of the indicators since the dimensions of some indicators are not the same or the data ranges are
quite different. The indicators that suggest that larger values are better and that smaller values are better are pre-processed by
Eq. (1) and Eq. (2), respectively:
$x_{ij} = \dfrac{x_{ij}^{o} - x_{j\,min}^{o}}{x_{j\,max}^{o} - x_{j\,min}^{o}}$          (1)
$x_{ij} = \dfrac{x_{j\,max}^{o} - x_{ij}^{o}}{x_{j\,max}^{o} - x_{j\,min}^{o}}$          (2)
where $x_{ij}^{o}(i = 1, \cdots, n; j = 1, \cdots, m)$ is the j-th indicator value of the i-th sample. $x_{j\,min}^{o}$ and $x_{j\,max}^{o}$ are the minimum and
maximum values of the j-th indicator, respectively. $x_{ij}$ is the normalized indicator value.





(2) The projection pursuit method projects high-dimensional data into one-dimensional linear space for research;
therefore, we construct a projection index function for linear projection. $a_j(j = 1, \cdots, m)$ is the projection direction, and
$z_i$ is the one-dimensional projection value of $x_{ij}$ which is defined as follows:
$z_i = \sum_{j=1}^{m} a_j x_{ij}$ (3)
(3) The projection value is selected by constructing an objective function, and its scattering characteristics should be as
follows: local projection points should be as dense as possible and it is better to concentrate the points into several clusters. At
the same time, the overall projection points should be spread as much as possible. Therefore, the projection objective
function($Q_a$) is defined as follow:
$Q_a = S_z D_z$ (4)
where $S_z$ is the standard deviation of the projection value $z_i$ (Pearson,1900); $D_z$ is the local density of the projection
value $z_i$. $S_z$ and $D_z$ are defined as follow:
$S_z = \sqrt{\dfrac{\sum_{i=1}^{n}(z_i - E_z)^2}{n-1}}$ (5)
$D_z = \sum_{i=1}^{n}\sum_{j=1}^{n}(R - r_{ij}) \cdot u(R - r_{ij})$ (6)
where $E_z$ is the average of all projection values $z_i(i = 1, \cdots, n)$ and $R$ is the window density of the local density. Through
experiments, it was found that using $0.1 S_z$ as the value of $R$ could ensure that the average number of projection points
contained in the window was not too small, which could make the deviation in the sliding average as small as possible. At the
same time, it could also prevent the value of $R$ from increasing too much as n increased; $r_{ij}$ is the distance between
projection values(i.e., $r_{ij} = |z_i - z_j|$); $u_t$ is unit step function that equals 0 if $t < 0$ or equals 1 if $t \geq 0$.
(4) For the projection objective function($Q_a$),its function value changes when the projection direction ($a_j$) changes. To
obtain an optimal projection direction, and ensure that the structural features of the high-dimensional data are displayed as
much as possible, the maximum value of the projection objective function should be solved, therefore, it is an optimization
problem:
$\text{Max}Q_a = S_z D_z$ (7)
$\text{s.t.} \sum_{j=1}^{m} a_j^2 = 1$ (8)
It is very difficult to solve this complicated nonlinear optimization problem by using thetraditional optimization method.
Therefore, the real-coded accelerated genetic algorithm (RAGA) was used to address this problem, and the RAGA, which
simulates the survival of the fittest and the intragroup chromosome information exchange mechanism, is a general global
optimization method (Yang et al., 2005.)



**3.2 Evaluation method for the hydrological alteration degree**
The IHA system, consisting of 33 hydrological indicators, is employed to assess hydrological alteration. The 33 IHAs are
categorized into five groups addressing the magnitude, timing, frequency, duration, and rate of change (Shiau and Wu., 2010)
and each group has a different ecological significance(Table 2).For the IHA statistics of the pre-impact period, its range of
variation between the 75th and 25th percentiles is considered as the ecological target range. The alteration degree ($D_i$) of the
post-impact flow regime for the i-th IHA parameter is calculated as follows:
$D_i = \left| \frac{N_{\text{oi}} - N_e}{N_e} \right| \times 100\%$ (9)
where $N_{\text{oi}}$ and $N_e$ are the observed and expected number of years during which the "post-impact" values of the i-th IHA
parameters should fall within the ecological target range, respectively. Ranges of 0-33%, 33%-66%, and 66%-100% are defined
as the evaluation boundaries of low, medium, and high alteration degrees, respectively. Then, the overall hydrological alteration
degree is calculated as follows:
$D = \frac{1}{m} \sum_{i=1}^{m} D_i$ (10)
where m is the number of parameters. In this study, 32 parameters were considered since there was no zero-flow day. As seen
from equation (10), the conventional method gives the same weight for each IHA parameter.
**3.3 FDC method and LOR analysis**
A flow duration curve is simply a plot of the ordered daily streamflow observations Q(i) (where i = 1 is the largest flow) as a
function of their exceedance probability ($p_i$) (Vogel et al.,2007)and is defined as follows:
$p_i = \frac{i}{n+1}$ (11)
where n is the number of flow days and i is the rank. In this study, two typical annual FDCs during the pre-impact period (the
25th percentile FDC and the 75th percentile FDC) were used as the comparison objects.
The length of record (LOR) method is used to provide quantitative advice on the length of record required for each IHA
parameter. The result of the LOR for a station is considered as a reference for other stations with similar hydrologic regimes;
therefore, the station with the smallest anthropogenic impact and longest record length in the study area was chosen for LOR
analysis. For each IHA parameter, we calculate its statistics for each year in a data set along with the long-term mean as the
reference for LOR analysis. Then, the statistics of each parameter are ordered randomly and grouped into record-length
increments ranging from two years to the full LOR. The mean of each increment is calculated for a comparison with the long-
term mean. This process is repeated 50,000 times, from which 95%, 90%, and 85% confidence intervals (CIs) are calculated.
Finally, we calculate the LOR required within 5% and 10% long-term mean errors at a specified confidence interval for the
river in the study. For convenience of discussion, the LOR result within 10% of the long-term mean with a CI of 85% is





abbreviated as 10/85 (Timpe and Kaplan, 2017).
**4.Results and discussion**
**4.1 Characteristic analysis of annual precipitation change**
When we evaluate the effect of reservoirs on the hydrological regime, it is necessary to consider the potential impacts of
climate change on hydrological data since here may be different climatic conditions in the pre-and post-stages (Wang et al.,
2017a). Therefore, the Mann–Kendall (hereafter referred to as MK) test method was used to analyze the trend in the annual
precipitation time series 1966-2017(52 years) for 28 stations in the Jinsha River basin. In the study area, annual precipitation
showed no significant trend or trend below the 10% significance level at 21 stations, and only 2 and 3 stations showed
increasing trends at 1% and 5% significance levels, respectively. A decreasing trend was found at only 1 station with 1%
significance level (Fig. 2). The moment estimation method was used to calculate the characteristic values of precipitation from
two short time series (1966-1998 and 1999-2017) and one long time series (1966-2017). Compared with the value of the long
time series, the relative errors of the mean of two short time series did not exceed ± 3% at 22 and 16 stations respectively, and
the absolute errors of coefficient of variation do not exceed ±0.03 at 26 and 18 stations. The results imply that the precipitation
series from 1999-2017(post-impact period) and 1966-1998(pre-impact period) have the same meteorological conditions, and
show no significant trends.
**4.2 Projection pursuit analysis**
**4.2.1 Applying data mining to identify indicator weights**
The IHA statistics software developed by the US Nature Conservancy (http://www.nature.org/) was used to analyze the daily
streamflow data of the Panzhihua, Huatan and Pingshan hydrological stations from 1966 to 2017, and a matrix of 32 IHA
statistics with long time series was obtained. According to the principle that the optimal pattern for a flow regime occurs in a
natural state with no interference (Bayley, 1995), that is, the characteristics of intra-annual cyclical changes in wet and dry
situations in a river are maintained, we first preprocessed the indicator system and transferred the high-dimensional data to
low-dimensional subspaces using the PP method, and then, obtained the optimal projection direction of each indicator by the
optimizing projection objective function and model parameters with the RAGA. The population number was 400, the
probability of crossover was 0.8, the number of excellent individuals was 20, and the acceleration time was 10 (Fig. 3). The
larger the optimal projection direction value is, the greater the contribution to the flow regime evaluation, that is, the higher
the weight of the indicator. As shown in Fig. 3, at the Pingshan and Huatan stations, the weight allocations are similar(Fig. 3b
and 3c), and the parameters with similar high weight values (greater than 0.04) are mean flow in January, April, May, annual



minimum discharge(one-day mean, seven-day mean, 90-day mean), base flow index, duration of high flow pulse, rise rate, fall
rate, and number of reversals, while at Panzhihua station(Fig. 3a), the high weight parameters are mean flow in January,
February, March, June, September, November, annual minimum discharge(three-day mean and thirty-day mean), base flow
index, date of minimum, count and duration of high pulse(Table 3). This result indicates that the data structure of the
characteristics of the flow regime characterized by IHA parameters has both similar and different parts upstream and
downstream, and it also implies that different weights may be related to tributary imports, reservoir constructions or interval
water supplies.
**4.2.2 Projection value calculation**
The projection values of the flow regime from the 1966-2017 hydrological series (Fig. 4) were obtained by substituting the
optimal projection directions into formula (3) at three stations. The results of the trend analysis on the projection values by the
MK test suggest that at Panzhihua station, the projection values fluctuated periodically without any significant change
trend(Fig. 4a), while the significant decreasing trends were found at the 1% significance level at the Huatan and Pingshan
stations (Fig. 4b and 4c), especially at Pingshan station, where the decreasing trend was more intense after 2012. The projection
value is a comprehensive evaluation result of flow regime changes and takes into account the monthly flow condition, the
magnitude and duration of extreme discharge conditions, the occurrence time of extreme discharge conditions, the frequency
and duration of high/low flow pulse, and the frequency and rate of hydrological process changes. The larger the projection
value is, the more distinct the intra-annual cyclical change in wet and dry situations, and the smaller the interference caused
by human activities. As shown in Fig. 4b and 4c, the projection values both began to show a significant decline in 1999, and a
more significant decreasing trend was found at Pingshan station from 2013 to 2017(Fig. 4c). The timing of the two drastic
changes coincides with the time when the Ertan Reservoir in the Yalong River(tributary) and the Xiangjiaba and Xiluodu
Reservoirs in the lower reaches of the Jinsha River (main stream) were first put into operation. This finding also implies that
the impact of giant reservoirs on the flow regime is substantial, and that the degree of impact is further aggravated with the
continuous construction of reservoirs.
**4.2.3 Evaluation of the hydrological alteration degree with a revised method**
According to the commissioning time of the first generator set in the Ertan Reservoir, the period of 1966-1998 is considered
as the natural state hydrological series unaffected by human activities(pre-impact period), and the period of 1999-2017 is
considered as the series affected by human activities(post-impact period).The alterations and weights of the 32 IHA parameters
are shown in Table 4. As shown in Table 4, the alteration degree calculated by the revised IHA method is larger than that by
the traditional method. For the Panzhihua, Huatan and Pingshan stations, the overall alteration degrees calculated by the revised



method are 0.29, 0.57, 0.54, and those by the traditional method are 0.28, 0.50 ,0.49 by the traditional method with relative
changes 3.57%, 14%, and 10.20%, respectively. The traditional IHA method, analyzing overall hydrological alteration with
the same weight for each parameter, constantly underestimates or overestimates actual hydrological changes since many
parameters are intercorrelated (Yang et al., 2019). Fig. 5 illustrates boxplots of the correlation coefficients between each IHA
parameter and the remaining 31 IHA parameters at the three hydrological stations mentioned above. At the Panzhihua station,
the absolute values of the correlation coefficients among the IHA parameters range from 0.0016 to 0.9976, with a mean of
0.2852(Fig. 5a). The absolute values of the correlation coefficients at the Huatan station range from 0.0 to 0.9876 with a mean
of 0.2931(Fig. 5b). The absolute values of the correlation coefficients at the Pingshan station range from 0.0012 to 0.9972 with
a mean of 0.2920(Fig. 5c). Fig. 5 shows that the correlations among parameters at the Huatan and Pingshan stations are stronger
than that at the Panzhihua station, which suggests that the correlations among parameters has an impact on the evaluation of
the hydrological alteration when combined with the results of the above two methods. The stronger the correlation among
parameters, the greater the impact is. The Panzhihua station is located 10 km upstream of the junction of the Yalong River and
the Jinsha River; therefore, operation of the Ertan Reservoir does not affect its hydrological streamflow series. This is also
confirmed by the fact that the overall hydrological alteration at the Panzhihua station is low.
**4.3 Cumulative impacts of cascade reservoirs on the flow regime**
**4.3.1 Hydrological alteration degree**
The large reservoirs of the Ertan, Xiluodu and Xiangjiaba were successively built along the Jinsha River. Three periods were
utilized for studying the cumulative impacts of cascade reservoirs. The Ertan Reservoir was put into operation in 1999, and
the Xiluodu and Xiangjiaba Reservoirs were both put into operation in 2013. Therefore, the first period is 1966-1998 with
natural flow regime conditions; the second period is 1999-2012 with the effects of individual reservoir; and the third period is
2013-2017 with the effects of the cascade reservoirs of Ertan, Xiluodu and Xiangjiaba. A total of 32 IHA statistics at the
Pingshan station were calculated, and the weights of each parameter were obtained by PP and RAGA. For the three different
periods, the alteration degrees of each parameter and the overall alteration degrees are shown in Table 5 and Table 6,
respectively. The cumulative impacts on the flow regime are very obvious with the successive construction of the reservoirs.
During the period of 1999-2012, the number of high alteration degree parameters was eight, with an overall alteration degree
of 47%, while during the period of 2013-2017, the number of high alteration degree parameters increased to thirteen, with an
overall alteration degree of 70%. In particular, for the parameters of the mean flow in May(5), base flow index(23), and low
pulse count(26), the alteration degrees of the three parameters were low during the period of 1999-2012, but they became high
alteration degrees during the period of 2013-2017.The increasing trends in the alteration degree are shown in the group 1, 2,





3, and 4. Only group 5 showed a decreasing trend in alteration degree. Overall, with the successive construction of the
reservoirs, the averaging effect of runoff became more obvious; that is, the flow reduced in the flood season, while it increased
in the dry season; the maximum value decreased; and the minimum value increased. As shown in Table 7, compared with the
period of 1966-1998, the changes in winter precipitation during the periods of 1999-2012 and 2013-2017 were very small, -
1.5% and 1.6%, respectively, but the minimum flow values of the two periods increased by approximately 11-25% and 30-
38%, respectively. However, slightly different changes in the flow regime were found in summer. Precipitation in summer had
slight increases during the two periods, 2.5% and 2.7%, respectively. Significant decreases in maximum flow values were
found during the period of 2013-2017, approximately 17-24%. However, during the period of 1999-2012, the maximum flow
values increased by 4%-8%, which is basically consistent with the increase in precipitation. These findings suggest that during
the period of 1999-2012, only the Ertan Reservoir was operating in the tributary; therefore, its ability to control runoff and the
impact on hydrological conditions on the outlet section of the basin were limited. In summer, the changes in the flow regime
were mainly influenced by precipitation. However, in winter, the regulation of the Ertan Reservoir was still relatively obvious.
With the operation of the Xiluodu and Xiangjiaba Reservoirs in the main-stream, the impacts of reservoir regulation on the
flow regime were greater in summer than before.
**4.3.2 FDC analysis**
To better compare and analyze the cumulative effect of cascade reservoirs on the flow regime, two years were selected at
Pingshan station, that is 2004 and 2016, based on the annual flow of periods of 1999-2012 and 2013-2017(Fig. 6). These years
represent the years with an annual flow in the 50th percentile, and their annual mean discharges are 4796 m³/s and 4063 m³/s,
respectively. Compared with the annual FDC above the 20th percentile in 2016, the high flow in 2004 was larger. The high
flow above the 20th percentile of the FDC typically occurred in summer (Gao et al., 2012), and the precipitation anomalies in
summer were 0 mm in 2004 and -29 mm in 2016. The summer runoff was basically consistent with precipitation in the same
period, indicating that summer runoff changes are mainly caused by seasonal precipitation changes. By analyzing the annual
FDCs between the 20th and 60th percentiles, we found that the flow in 2016 was smaller than that in 2004. This flow occurred
mostly in fall, and the fall precipitation anomalies were -3mm in 2004 and -47mm in 2016. We analyzed the consistency of
the underlying surface in the two pre- and post-impact periods, and the results indicate that the precipitation–runoff correlation
trend lines for the two periods nearly coincide (Fig. 7). Therefore, the main reason for the phenomenon in fall is the
accumulation of the impact of reservoir storage after the Xiangjiaba and Xiluodu Reservoirs were put into operation. The
cumulative impact was much larger than the single impact of the Ertan Reservoir before 2012. At the same time, even if there
was a large increase in precipitation in fall of 2016, the impact of precipitation on the flow was also weakened by cumulative
impact of water storage. The low flows below the 80th percentile of the FDC occurred in winter, and two typical annual FDCs





during the pre-impact period (the 25th percentile FDC and the 75th percentile FDC) were almost coincident during the low
flows. This indicates that the changes in runoff in the dry season were slight during the pre-impact period. The winter
precipitation anomalies were -3 mm in 2004 and 3 mm in 2016.Therefore, the reservoir water release was the main reason that
the annual FDCs below the 80th percentile in 2004 and 2016 were generally above two typical annual FDCs. Due to the
cumulative effect, the low flow in 2016 was even higher than the low flow in 2004.

**4.4 Data reliability analysis**

The characterization of natural and altered flow regimes using IHA requires adequate flow data (Zhang et al., 2018). Richter
et al. (1997) suggested using >20 years of pre- and post-impact data to characterize the hydrologic regime. Timpe and Kaplan
(2017) found that fewer than 20 years of data could be used to yield statistically significant IHA results for a number of rivers
across the Amazon by using the length of record (LOR) method. Given this uncertainty, further research is needed to determine
the reliability of the data required by IHA. We chose 47 years from 1952 to 1998 at the Huatan station as the LOR calculation
period with the smallest anthropogenic impact and the longest record length in the study area. Table 8 shows the length of
record required for the 32 IHA parameters within 5% and 10% long-term mean errors at different specified confidence intervals
at the Huatan station. Comparing the results between different groups, it is observed that the data volume requirement in group
4 is the highest, while when comparing and analyzing within the same group, it is observed that the amount of data required
to describe the parameters for low flow is less than that required to describe the parameters for a relatively high flow. For
example, the amount of data required for monthly mean flow in the flood season is higher than that for the monthly mean flow
in the dry season in group 1. Zhang et al. (2017) found that the amount of data required has a consistent relationship with the
amount of average monthly flow and the variability in hydrological data. Both the Huatan and Pingshan stations are located
downstream of the Jinsha River with similar hydrologic regimes (Fig.3). Referring to the results of Table 8, the 33-year daily
streamflow data from 1966 to 1998 at the Pingshan station fully satisfy the highest requirement (31 years) to produce a 10/85
LOR result for all parameters. These data also satisfy the requirement to produce a 10/90 LOR result except that the parameter
of high pulse duration requires 34 years for analysis. Furthermore, the number of the IHA parameters that satisfy the
requirement to produce 10/95, 5/85, 5/90, and 5/95 LOR results are 30, 30, 28, and 28, respectively. This indicates that the 33-
years daily streamflow data at the Pingshan station could capture intra- and interannual flow variations. At the same time, the
19 years of post-impact data we collected from 1999 to 2017 could mainly satisfy the data requirement for analysis. Therefore,
the data collection in this study basically satisfies the requirements for the analysis, and the overall evaluation of the
hydrological alteration degree is basically reasonable.





**4.5 Attempts at ecological regulation**
Li et al., (2006) found that the total number of days with rising water from May to June in the Yangtze River is an important
environmental driving factor that determines the amount of spawn produced by fish with pelagic eggs. Four major Chinese
carps, i.e., black carp (Mylopharyngodon piceus), grass carp (Ctenopharyngodon idellus), silver carp (Hypophthalmichthys
molitrix) and big head carp (Aristichthysmobilis) are typical fish with the pelagic eggs and the most widely distributed species
in protected areas. The cumulative impact of the construction and operation of cascade reservoirs on the flow regime in the
downstream nature reserve of the Jinsha River has aroused widespread concern. Through analysis, the average annual rise rate
at Pingshan station in 1966-1998 is 132m$^3$/s. During the period from May 15 to May 18, 2018, the management institution
conducted a joint eco-hydrological regulation test of the Xiluodu-Xiangjiaba Reservoirs for 4 days (Fig. 8). On May 15, the
outflow discharge of the Xiluodu Reservoir was 2770m$^3$/s, and it increased to 3420m$^3$/s on May 16. After that, it increased
slightly to 3470m$^3$/s in the next two days. On May 14, the outflow discharge of the Xiangjiaba Reservoir was 2740m$^3$/s, and
it increased to 3330m$^3$/s on May 15. Then, it evenly increased 300m$^3$/s every day and reached 4320m$^3$/s. Ecological regulation
promoted the spawning of fish. The Yibin monitoring section showed obvious spawning during the ecological regulation period,
and fish spawning under the reservoir was stimulated obviously. The spawning peak was found at the Jiangjin monitoring
section on the 6th day after the end of the regulation. Before the regulation, the amount of spawning was much lower in the
Jiangjin monitoring section, but a significant increasing trend was found during the regulation and after the regulation. During
the period of joint regulation, the number of fish eggs was 3 million in the Yinbin section, 7 million in the Luzhou section, and
173 million in the Jiangjin section. Four major Chinese carps have a positive response to the flood peak process caused by
manual regulation, and the joint eco-hydrological regulation has a good effect on the natural reproduction of these four major
Chinese carps. In addition to the natural reproduction of the four major Chinese carps, the feasibility of studying for other
important species is still in progress.
**5 Conclusions**
In this study, a revised IHA method is presented by combining projection pursuit (PP) with a real-coded genetic algorithm
(RAGA) to obtain the weight of each IHA parameter. The method is applied to assess the cumulative impacts of cascade
reservoirs on the flow regime in the Jinsha River. The main points can be summarized as follows:
(1) The impacts of the construction and operation of the cascade reservoirs on the flow regime are huge. Using the revised
IHA method to analyze the cumulative effects of Ertan, Xiangjiaba and Xiluodu Reservoirs on the flow regime of the
outlet section of the Jinsha River, we found that with the continuous construction of the reservoir, the alteration degrees
of IHA parameters in Groups 1, 2, 3, and 4 are gradually increasing, but are decreasing in Group 5(rise rate, fall rate,
number of reversals). Due to reservoir water storage and release, the FDC shows decreasing trends in high flow, and





increasing trends in low flow. The whole curve shows the characteristics of a head drop and the tail lift. The maximum
flow is reduced, and the minimum flow is increased. The rate and frequency of discharge changes tend to be subtle. As
the cascade reservoirs are completed, the flow regime alteration at the outlet section is more stable. This change has a
negative impact on downstream fish reproduction and ecological protection.
(2) The traditional IHA method employs 33 parameters to quantify the characteristics of flow regime changes, and analyze
the overall hydrological alteration with the same weight for each parameter. The revised IHA method gives each
parameter its own weight by applying a projection pursuit model to project high-dimensional data into a low-dimensional
space and optimizing the projection direction of each parameter. This method achieves a more reasonable evaluation of
hydrological alterations and overcomes the problem of underestimating the hydrological alterations in the study area due
to the difference in the degree of alteration and the intercorrelation among IHA parameters.
(3) Previous studies have suggested using >20 years of pre- and post-impact data to characterize the hydrologic regime. In
this study, we chose 47 years from 1952 to 1998 at the Huatan station as the LOR calculation period. As a reference, the
33-years daily streamflow data from 1966 to 1998 at Pingshan station fully satisfy the highest requirement (31 years) to
produce a 10/85 LOR result. These data also satisfy the requirement to produce a 10/90 LOR result except that the
parameter of high pulse duration requires 34 years for analysis. In summary, the data collected in this study basically
satisfy the requirements for the analysis, and the overall hydrological alteration degree evaluated by the IHA method is
essentially reliable and reasonable.
**Author contribution**
Xingyu Zhou and Xiaorong Huang suggested the idea and formulated the overarching research goals and aims. Xingyu Zhou,
Hongbin Zhao and Kai Ma corrected and managed the data. Xingyu Zhou and Xiaorong Huang employed statistical method
to analyze study data. Xingyu Zhou prepared the manuscript with contributions from all co-authors
**Acknowledgements**
This research was jointly supported by the National Natural Science Foundation of China (Grant No. 51579161,51779160)
and the China Meteorological Data Sharing Service System. The opinions expressed here are those of the authors, and not
those of other individuals or organizations.
**Competing interests**
The authors declare that they have no conflict of interest.





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





Table 1. Large reservoirs built in the Jinsha River Basin

| Reservoir | Total storage capacity ($10^8$m$^3$) | Regulating storage ($10^8$m$^3$) | Installed capacity ($10^4$kW) | Annual energy production ($10^8$ kW·h) | Pool level (m) | Basin area ($10^4$km$^2$) | First impoundment year |
|---|---|---|---|---|---|---|---|
| Ertan | 58 | 33.7 | 330 | 170 | 240 | 11.64 | 1999 |
| Xiluodu | 126.7 | 64.6 | 1260 | 573.5 | 600 | 45.33 | 2013 |
| Xiangjiaba | 51.63 | 9.03 | 600 | 307.47 | 384 | 45.88 | 2013 |





Table 2. List of hydrological stations and their features.

| Station | Longitude (E) | Latitude(N) | Drainage area $(10^4 km^2)$ | Annual discharge $(10^8 m^3)$ | Runoff data | |
|---|---|---|---|---|---|---|
| | | | | | Record period | Length(year) |
| Panzhihua | 101°44′41″ | 26°38′19″ | 25.92 | 561.38 | 1966-2017 | 52 |
| Huatan | 102°54′03″ | 26°59′45″ | 42.59 | 1255.11 | 1952-2017 | 66 |
| Pingshan | 104°15′51″ | 28°37′29″ | 45.85 | 1426.30 | 1966-2012 | 47 |
| Xiangjiaba | 104°24′29″ | 28°38′00″ | 45.88 | 1286.00 | 2013-2017 | 5 |

(The Xiangjiaba station was also called the Pingshan Station in this study.)





Table 3. 33 Indicators of Hydrologic Alteration

|  | Group 1 |  | Group 2 |  | Group 3 |
|---|---|---|---|---|---|
| 1 | Mean flow in January | 13 | 1-day minimum | 24 | Date of minimum |
| 2 | Mean flow in February | 14 | 3-day minimum | 25 | Date of maximum |
| 3 | Mean flow in March | 15 | 7-day minimum |  | Group 4 |
| 4 | Mean flow in April | 16 | 30-day minimum | 26 | Low pulse count |
| 5 | Mean flow in May | 17 | 90-day minimum | 27 | Low pulse duration |
| 6 | Mean flow in June | 18 | 1-day maximum | 28 | High pulse count |
| 7 | Mean flow in July | 19 | 3-day maximum | 29 | High pulse duration |
| 8 | Mean flow in August | 20 | 7-day maximum |  | Group 5 |
| 9 | Mean flow in September | 21 | 30-day maximum | 30 | Rise rate |
| 10 | Mean flow in October | 22 | 90-day maximum | 31 | Fall rate |
| 11 | Mean flow in November | 23 | Base flow index | 32 | Number of reversals |
| 12 | Mean flow in December | 33 | Zero flow day* |  |  |

*This parameter is excluded from the study





Table 4. Alteration degree and weight of 32 IHA parameters

| Parameter | Panzhihua Station (1966-2017) | | Huatan Station (1966-2017) | | Pinshan Station (1966-2017) | |
|---|---|---|---|---|---|---|
| | Alteration Degree | Weight | Alteration Degree | Weight | Alteration Degree | Weight |
| IHA Group 1 | | | | | | |
| 1 | 5% | **0.0513** | 89% | **0.0529** | 79% | **0.0519** |
| 2 | 16% | **0.0566** | 100% | 0.026 | 79% | 0.0251 |
| 3 | 68% | **0.0458** | 100% | 0.0202 | 89% | 0.0181 |
| 4 | 5% | 0.0387 | 47% | **0.0486** | 58% | **0.0501** |
| 5 | 26% | 0.003 | 68% | **0.0445** | 37% | **0.0442** |
| 6 | 5% | **0.0492** | 5% | 0.0113 | 5% | 0.0157 |
| 7 | 16% | **0.0403** | 26% | 0.0291 | 47% | 0.0303 |
| 8 | 16% | 0.0042 | 26% | 0.0221 | 5% | 0.0201 |
| 9 | 26% | **0.0572** | 5% | 0.0425 | 5% | 0.0393 |
| 10 | 37% | 0.008 | 58% | 0.0193 | 58% | 0.0194 |
| 11 | 26% | **0.0456** | 5% | 0.0133 | 16% | 0.013 |
| 12 | 16% | 0.0142 | 16% | 0.0385 | 16% | 0.0352 |
| IHA Group 2 | | | | | | |
| 13 | 79% | 0.0293 | 79% | **0.0571** | 58% | **0.0563** |
| 14 | 79% | **0.0459** | 89% | 0.039 | 68% | 0.0376 |
| 15 | 68% | 0.0123 | 68% | **0.0608** | 89% | **0.0598** |
| 16 | 47% | **0.051** | 89% | 0.0311 | 100% | 0.0287 |
| 17 | 47% | 0.0395 | 89% | **0.0424** | 89% | **0.0456** |
| 18 | 5% | 0.0268 | 16% | 0.0218 | 16% | 0.0226 |
| 19 | 5% | 0.0374 | 16% | 0.0175 | 16% | 0.0183 |
| 20 | 5% | 0.0234 | 5% | 0.0008 | 26% | 0.0025 |
| 21 | 16% | 0.0262 | 26% | 0.0092 | 47% | 0.0114 |
| 22 | 47% | 0.0281 | 5% | 0.0142 | 16% | 0.0164 |
| 23 | 26% | **0.0444** | 16% | **0.0587** | 37% | **0.0621** |
| IHA Group 3 | | | | | | |
| 24 | 16% | 0.0006 | 68% | 0.0137 | 58% | 0.0132 |
| 25 | 5% | **0.0503** | 26% | 0.0396 | 5% | **0.0432** |
| IHA Group 4 | | | | | | |
| 26 | 26% | 0.0124 | 68% | 0.008 | 47% | 0.0052 |
| 27 | 16% | 0.0042 | 100% | 0.0273 | 89% | 0.0274 |
| 28 | 5% | **0.0548** | 5% | 0.011 | 47% | 0.0082 |
| 29 | 68% | **0.0427** | 26% | **0.0432** | 26% | **0.0446** |
| IHA Group 5 | | | | | | |
| 30 | 16% | 0.0136 | 68% | **0.0492** | 37% | **0.0454** |
| 31 | 16% | 0.0131 | 100% | **0.0474** | 100% | **0.0449** |
| 32 | 37% | 0.0299 | 100% | **0.0399** | 100% | **0.0442** |
| mean value | 28% | | 50% | | 49% | |
| Weighted mean value | 29% | | 57% | | 54% | |

The values in bold mean high weights.





Table 5. Alteration degree of 32 indicators in Pingshan station in different periods

| Parameter | Pingshan Station 1999-2012 | Pingshan Station 2013-2017 | Parameter | Pingshan Station 1999-2012 | Pingshan Station 2013-2017 |
|---|---|---|---|---|---|
| | Alteration Degree | Alteration Degree | | Alteration Degree | Alteration Degree |
| 1 | 71% | 100% | 17 | 86% | 100% |
| 2 | 86% | 60% | 18 | 0% | 60% |
| 3 | 86% | 100% | 19 | 0% | 60% |
| 4 | 43% | 100% | 20 | 14% | 60% |
| 5 | 14% | 100% | 21 | 43% | 60% |
| 6 | 14% | 20% | 22 | 0% | 60% |
| 7 | 43% | 60% | 23 | 14% | 100% |
| 8 | 0% | 20% | 24 | 57% | 60% |
| 9 | 14% | 20% | 25 | 14% | 60% |
| 10 | 57% | 60% | 26 | 29% | 100% |
| 11 | 0% | 60% | 27 | 86% | 100% |
| 12 | 0% | 60% | 28 | 57% | 20% |
| 13 | 43% | 100% | 29 | 14% | 60% |
| 14 | 57% | 100% | 30 | 57% | 20% |
| 15 | 86% | 100% | 31 | 100% | 100% |
| 16 | 100% | 100% | 32 | 100% | 100% |





Table 6. Overall degree of alteration of five groups of IHA parameters

|  | Pingshan Station 1998-2012 | Pingshan Station 2013-2017 |
|---|---|---|
| Group 1 | 16% | 23% |
| Group 2 | 20% | 28% |
| Group 3 | 1% | 4% |
| Group 4 | 1% | 10% |
| Group 5 | 9% | 5% |
| Overall degree of alteration | 47% | 70% |

(The weight of each parameter has been considered)



Table 7. Changes in the annual minimum and annual maximum flows
between the periods 1966-1998, 1999-2012 and 2013-2017

| Indicator | Pre-impact period 1966-1998 | Post--impact period 1999-2012 | Relative changes (%) | Post-impact period 2013-2017 | Relative changes (%) |
|---|---|---|---|---|---|
| Precipitation in winter ( mm ) | 12.3 | 12.1 | -1.5% | 12.5 | 1.6% |
| Precipitation in summer ( mm ) | 394.5 | 404.3 | 2.5% | 405.0 | 2.7% |
| 1-day minimum flow(m$^3$/s) | 1216 | 1349 | 11% | 1588 | 31% |
| 3-day minimum flow(m$^3$/s) | 1221 | 1393 | 14% | 1591 | 30% |
| 7-day minimum flow(m$^3$/s) | 1231 | 1446 | 17% | 1605 | 30% |
| 30-day minimum flow(m$^3$/s) | 1269 | 1589 | 25% | 1753 | 38% |
| 1-day maximum flow(m$^3$/s) | 16525 | 17150 | 4% | 12580 | -24% |
| 3-day maximum flow(m$^3$/s) | 16000 | 16784 | 5% | 12458 | -22% |
| 7-day maximum flow(m$^3$/s) | 14915 | 15854 | 6% | 11816 | -21% |
| 30-day maximum flow(m$^3$/s) | 11889 | 12839 | 8% | 9835 | -17% |



Table 8.Length of record(LOR)results for each IHA parameter

| IHA Group | parameter | LOR results(years) | | | | | |
|---|---|---|---|---|---|---|---|
| | | 5/95 | 5/90 | 5/85 | 10/95 | 10/90 | 10/85 |
| Group1 | January | 15 | 11 | 8 | 5 | 3 | 2 |
| | February | 14 | 10 | 7 | 5 | 3 | 2 |
| | March | 14 | 10 | 7 | 5 | 3 | 2 |
| | April | 16 | 11 | 8 | 5 | 3 | 2 |
| | May | 19 | 14 | 10 | 7 | 5 | 3 |
| | June | 29 | 23 | 18 | 13 | 10 | 7 |
| | July | 32 | 26 | 21 | 16 | 11 | 8 |
| | August | 33 | 28 | 23 | 18 | 13 | 9 |
| | September | 26 | 20 | 16 | 12 | 8 | 6 |
| | October | 28 | 22 | 17 | 12 | 9 | 6 |
| | November | 20 | 15 | 12 | 7 | 5 | 3 |
| | December | 17 | 13 | 9 | 6 | 4 | 3 |
| Group2 | 1-day min | 14 | 10 | 7 | 5 | 3 | 2 |
| | 3-day min | 14 | 10 | 7 | 5 | 3 | 2 |
| | 7-day min | 14 | 10 | 7 | 5 | 3 | 2 |
| | 30-day min | 14 | 10 | 7 | 5 | 3 | 2 |
| | 90-day min | 14 | 10 | 7 | 5 | 3 | 2 |
| | 1-day max | 27 | 22 | 17 | 12 | 8 | 6 |
| | 3-day max | 27 | 22 | 17 | 12 | 8 | 6 |
| | 7-day max | 27 | 22 | 17 | 12 | 8 | 6 |
| | 30-day max | 26 | 20 | 16 | 12 | 8 | 6 |
| | 90-day max | 25 | 19 | 15 | 10 | 7 | 5 |
| | Base flow | 21 | 16 | 12 | 8 | 5 | 4 |
| Group3 | Date min | 27 | 23 | 20 | 15 | 10 | 7 |
| | Date max | 9 | 6 | 4 | 3 | 2 | 2 |
| Group4 | Lo pulse # | 41 | 35 | 33 | 27 | 22 | 18 |
| | Lo pulse L | 43 | 42 | 41 | 37 | 32 | 27 |
| | Hi pulse # | 40 | 38 | 35 | 27 | 20 | 16 |
| | Hi pulse L | 45 | 44 | 41 | 38 | 34 | 31 |
| Group5 | Rise rate | 31 | 25 | 21 | 16 | 12 | 9 |
| | Fall rate | 25 | 21 | 16 | 12 | 8 | 5 |
| | Reversals | 11 | 8 | 6 | 3 | 2 | 2 |


**Fig.1**

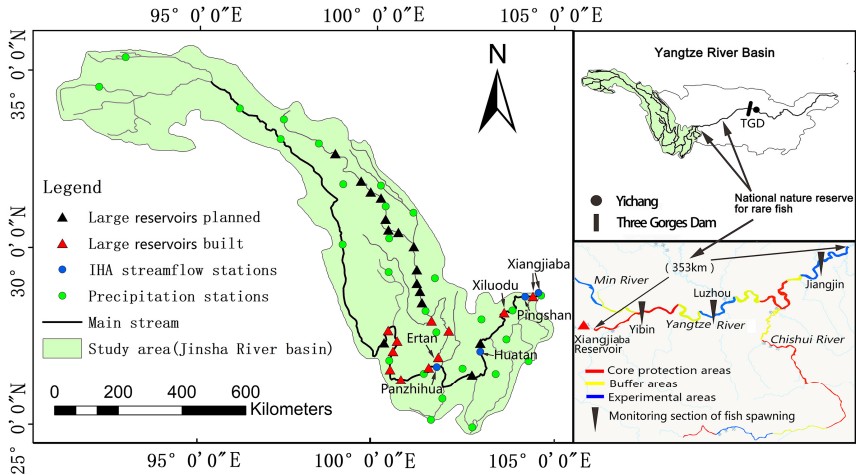





**Fig.2**

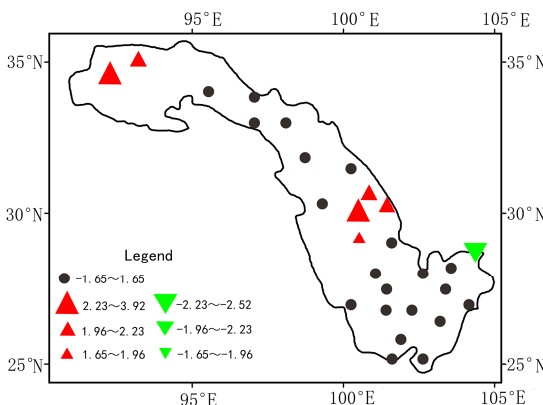





**Fig.3**

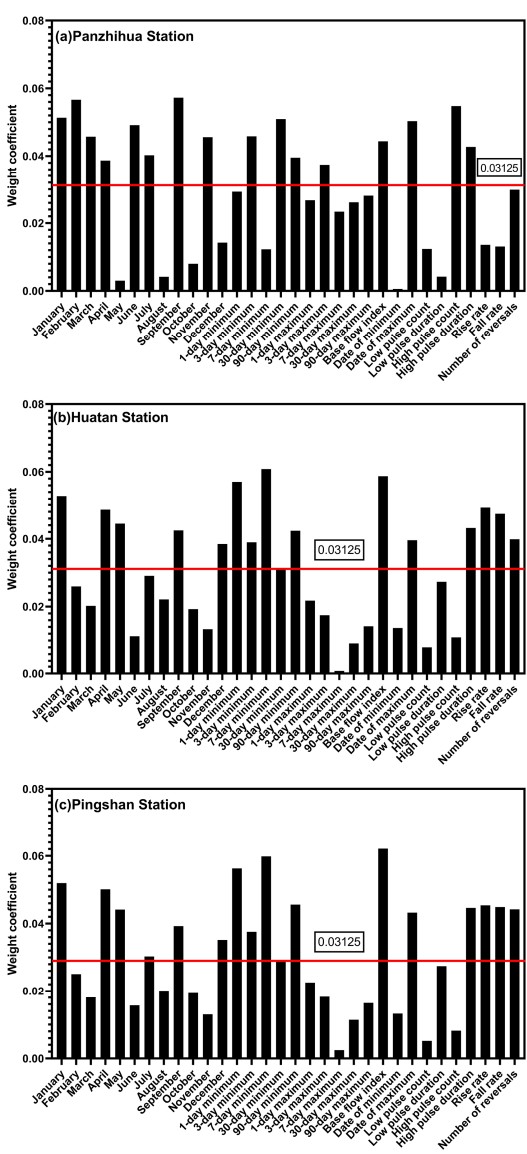





**Fig.4**

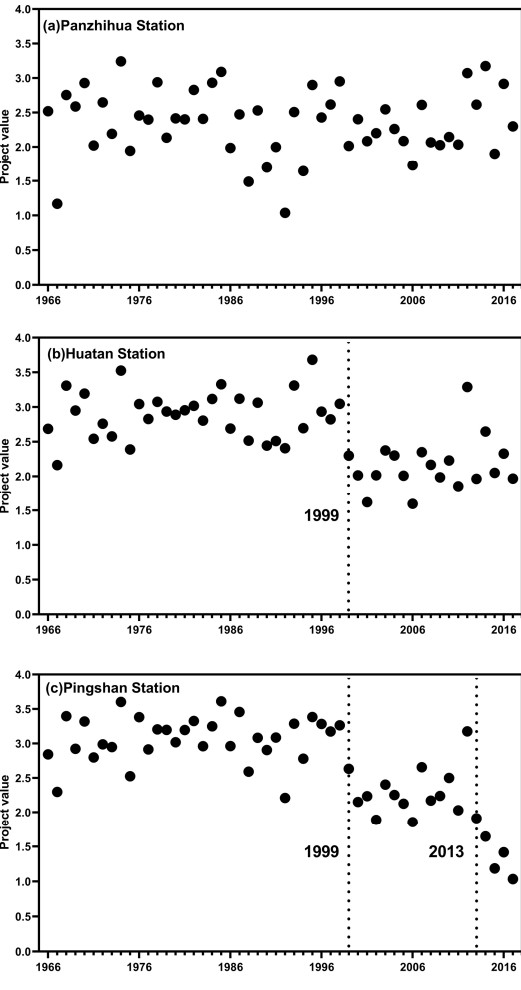



**Fig.5**

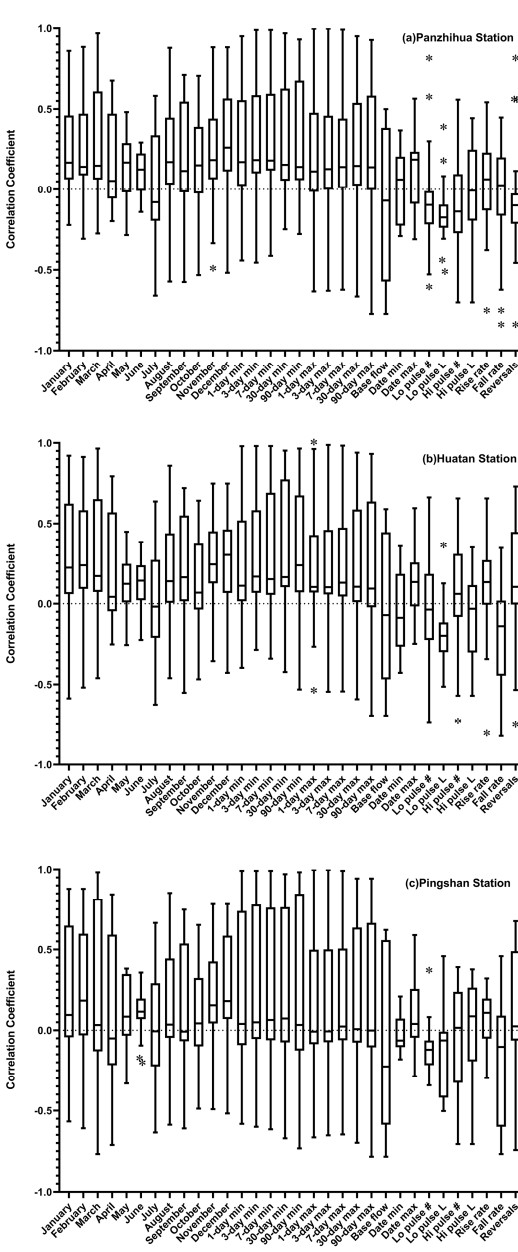



**Fig.6**

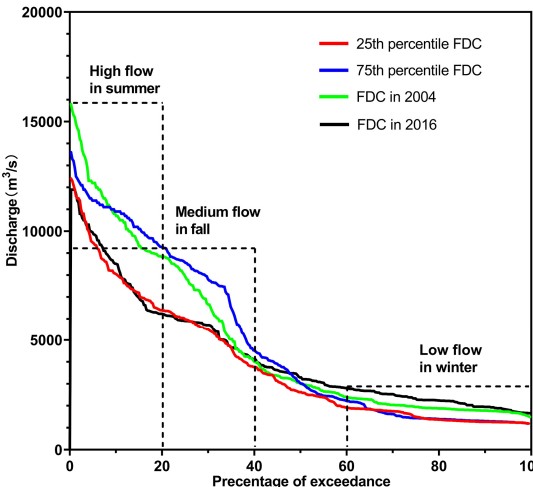





**Fig.7**

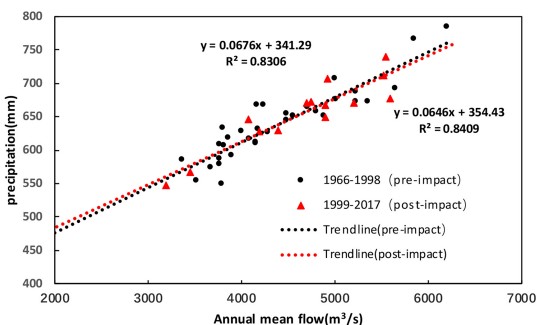



**Fig.8**

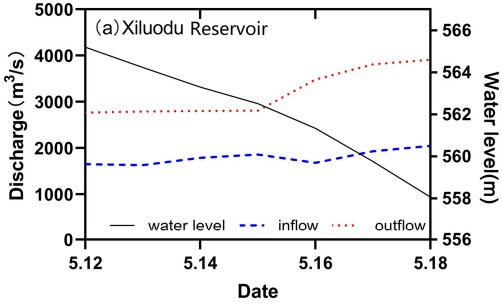
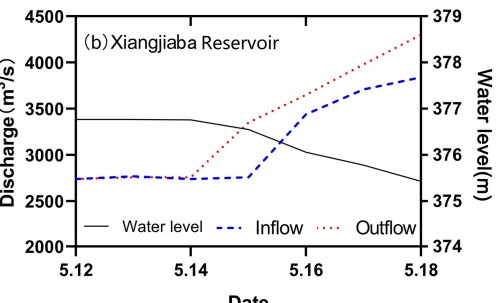



**figure captions**

Fig 1. Generalized map of study region.

Fig 2. Precipitation changes in the Jinsha River Basin:(a) Trend of annual Precipitation at 28 stations between 1966 and 2017(Upward (downward) triangles indicate positive (negative) trends from MK test. The size of the triangles depicts the significance levels 10% (small), 5% (medium), and 1% (large). Black dots show stations with no trends or trends below 10% significance level. Value in the legend is the standardized statistics Z value).

Fig 3. Value of weights of 32 IHA parameters in the (a) Panzhihua Station, (b) Huatan Station, and (c) Pingshan Station. The red line is equal weight line (1/32=0.03125).

Fig 4. Project values in the (a) Panzhihua Station, (b) Huatan Station, and (c) Pingshan Station.

Fig 5. Correlation coefficients among the IHA statistics for the observed data sets in the (a) Panzhihua Station, (b) Huatan Station, and (c) Pingshan Station. (* means outlier.)

Fig. 6 Annual flow duration curves in 2004 and 2016.

Fig. 7 Correlation between runoff and precipitation for the periods of pre– and post–impact.

Fig 8. Hydrograph for daily average inflow, outflow and reservoir water level in the (a) Xiluodu and (b) Xiangjiaba Reservoirs.