# Peer review of "Development of a revised IHA method for analysing the cumulative impacts of cascading reservoirs on flow regime"

_Hydrology and Earth System Sciences, 2019_

## Referee Comment (RC1) · Anonymous Referee #1 · 7 Mar 2020

The correlation of indicators has always been the focus of attention when using the IHA method to evaluate the degree of change in hydrological situation. In this script, the evaluation method was improved by using the projection pursuit method based on real-coded accelerated genetic algorithm, which gives different weights to the indicators, reduces the correlation among indicators, and makes the evaluation results more reasonable and scientific. The idea of this study is innovative, and many researchers have used the principal component method to analyze the correlation among indicators. What are the advantages of this method proposed compared with the principal component method?

---

## Referee Comment (RC2) · Anonymous Referee #2 · 11 Mar 2020

In this paper, by utilizing the projection pursuit (PP) and real-coded accelerated genetic algorithm (RAGA), the author proposed a revised IHA method to evaluate the cumulative impacts of cascading reservoirs on the flow regime. The research has positive significance for ecological reservoir operation and sustainable water resource management under future scenarios. Major Issues: In Results and discussion section, the paper presents the results for the Jinsha River Basin but the results of this study were not properly discussed based on the previous literature. Moreover, the content written in Section 4.5 only introduces the existing ecological regulation, which is not relevant to the research results and lacks sufficient discussion. It is suggested that the relevant content in Section 4 be polished, in order to make the expression of this part more reasonable and clear. Minor Issues: 1. In Method section, there is no application de-

scription of the real-coded accelerated genetic algorithm (RAGA) method using in this study, which is suggested to be supplemented. 2. In Section 4.1, the author stated that the Mann–Kendall test method was used to analyze the trend in the annual precipitation, but there is no relevant description of the method used in the study. It is recommended to add them in Method section. 3. Line 146: "The 33 IHAs are categorized into five groups addressing the magnitude, timing, frequency, duration, and rate of change (Shiau and Wu., 2010) and each group has a different ecological significance(Table 2)". Please check the original expression, it is recommended to replace Table 2 with Table 3. 4. The first row and column in Table 1 are all bold, but not in other tables. Please be consistent in the format of all tables. 5. It is recommended to add the name of related station in the titles of Figure 6, Figure 7, Table 7, and Table 8, so that when the readers read the figures and tables individually, they can also clearly understand the meaning of the authors.

---

## Referee Comment (RC3) · Anonymous Referee #3 · 16 Mar 2020

In this manuscript, the authors developed a revised IHA method by using the projection pursuit (PP) and real-coded accelerated genetic algorithm (RAGA). The data reliability was analyzed by using length of record (LOR) method. The cumulative impacts of cascade reservoirs on flow regime were analyzed in the Jinsha River Basin. It seems that the improved method has good reliability, and the authors also considered the impact of the data length on the evaluation results, making the evaluation results more scientific and reliable. ÂăIn the section of discussing the reliability of data length, we could see that the evaluation results of different data lengths showed obvious uncertainties. Therefore, does this uncertainty appear differently at different stages (pre- and post-impact periods) ?
* * *
[Figure]

694, 2020.

---

## Referee Comment (RC4) · Anonymous Referee #4 · 18 Mar 2020

Optimum operation cascade reservoirs is significant to imrpove the benefits of water management and ecological protection. The way to weight and balance various objectives is always the ctitical points in this field. The authers present a revised IHA method for cumulative impacts of cascade reservoirs on flow regime is proposed by using the Projection Pursuit (PP) method based on the Real-coded Accelerated Genetic Algorithm (RAGA). They analyzed the cumulative effects of the Ertan, Xiangjiaba and Xiluodu Reservoirs on the flow regime of the outlet section of the Jinsha River. The topic is interesting and the technical soundness of the method is Generally, the contents are valuable for researchers and managers ivoveled in the work. For better understanding of the methodology and application, some parts are still recommended to discussed and illustrated more, as follows: 1. section 2.1 could be simplified, Make

it concise please. 2. The m-k method used in this paper, which is the basis distinguishing method, please adding some distinguishing methods to make readers more convinced. 3. The cascade reservoirs are built, the flow regime would be altered and not suited for spawning. Please clarify how and to what extent the former will impact the latter in your study area. It can be accepted and publised after pertinent revision.

———————————————

---

## Referee Comment (RC5) · Anonymous Referee #5 · 18 Mar 2020

To date, many approaches have already been developed to quantify the extent of  ̂ a ̆hydrological alterations caused by reservoirs. Although efforts have addressed the relationship resulting in statistical redundancy, the inter-correlations have not been analyzed comprehensively still. A new method is proposed to eliminate self-correlations among the 33 parameters and provide beneficial insights into water resources management. In Table 6, the overall degree of alteration of group 1-4 is increased from 1998-2012 to 2013-2017, but the fifth group is decreased, please explain the reasons.

---

## Author Comment (AC3) · 19 Mar 2020

Firstly, we thank to reviewer very much. Aim at suggestions for our manuscript, we will try to enrich Results and discussion section, and improve section 4.5. Moreover, some minor Issues mentioned will be modified.

---

## Author Comment (AC4) · 19 Mar 2020

Thank you reviewer. Some minor issues mentioned will be modified. In order to convince the reader a greater degree, some other methods will be used.

---

## Author Comment (AC5) · 19 Mar 2020

Thank you, commentator. As we know, the impacts of the construction and operation of the cascade reservoirs on the flow regime are huge. Construction and regulation of reservoirs reduces flow in flood season, increases flow in dry season, and significantly alters the monthly discharge regimes. In general, the impacts on the flow regime are further aggravated with the continuous construction of reservoirs, the averaging effect of runoff became more obvious, therefore, the group 5 ( Rise Rate, Fall Rate and Number of Reversals) showed a decreasing trend in alteration degree during the period of 2013-2017.

[Figure]

694, 2020.

---

## Author Response (AR1)

Dear Editors and Reviewers:

Thank you for your letter and for the reviewers' comments concerning our manuscript entitled "Development of a revised IHA method for the cumulative impacts of cascade reservoirs on flow regime" (No.: hess-2019-694). Those comments are all valuable and very helpful for revising and improving our paper. We have studied comments carefully and have made correction which we hope meet with approval. The main corrections in the paper and the responds to reviewer's comments are as flowing:

Responds to reviewer's comments:

**Referee #1**

**1.** "The idea of this study is innovative, and many researchers have used the principal component method to analyze the correlation among indicators. What are the advantages of this method proposed compared with the principal component method?"

**Response:** The Principal Component Analysis is to find a way to replace the old variables with fewer new variables and keep as much information as possible, unfortunately, a small amount of information will be lost, which will affect the evaluation to some extent. The Projection Pursuit (PP) method is a technique of falling high dimension and the Real-coded Accelerated Genetic Algorithm (RAGA) is a method of optimum. Through combining the PP and RAGA, the high-dimensional data are easily changed into low- dimensional space and the characteristics of high-dimensional data can be summarized almost no loss of information.

Special thanks to you for your good comments.

**Referee #2**

**1.** "In Results and discussion section, the paper presents the results for the Jinsha River Basin but the results of this study were not properly discussed based on the previous literature."

**Response:** The relevant discussion has been supplemented from line 296 to 308.

**2.** "The content written in Section 4.5 only introduces the existing ecological regulation, which is not relevant to the research results and lacks sufficient discussion. It is suggested that the relevant content in Section 4 be polished"

**Response**: The discussion on the impact of reservoir construction on fish reproduction in May and the restoration of ecological regulation have been added in Section 4.5 from line 379 to 387.

**3.** "In Method section, there is no application description of the real-coded accelerated genetic algorithm (RAGA) method using in this study, which is suggested to be supplemented."

**Response:** The application description of RAGA method has been supplemented from Line 147 to Line 170.

**4.** "It is recommended to add the relevant description of the Mann–Kendall test method in Method section."

**Response:** The relevant description of the Mann–Kendall test method has been added from Line 202 to Line 212.

**5.** "Please check the original expression, it is recommended to replace Table 2 with Table 3 at line 146."

**Response:** We have made correction according to referee's comment at Line 174.

**6.** "The first row and column in Table 1 are all bold, but not in other tables. Please be consistent in the format of all tables"

**Response:** The format of all tables has been unified.

**7.** "It is recommended to add the name of related station in the titles of Figure 6, Figure 7, Table 7, and Table 8"

**Response**: The names of related station have been added in the titles.

    Special thanks to you for your good comments.

**Referee #3**

**1.** "In the section of discussing the reliability of data length, we could see that the evaluation results of different data lengths showed obvious uncertainties. Therefore, does this uncertainty appear differently at different stages (pre- and post-impact periods)?

**Response:** Richter et al. (1997) suggested collecting >20 years of pre- and post-impact flow data when using IHA to assess hydrological alteration. Otherwise, the interannual climate change would cause great interference in the statistical results. Therefore, in order to capture the characteristics of hydrological changes, a long series of data is necessary. During the pre-impact period, the uncertainty is mainly caused by climate change. With the continuous construction of dams, the uncertainty is jointly determined by dam operation and climate change simultaneously, the former usually play import role in altering the flow regime.

    Special thanks to you for your good comments.

**Referee #4**

**1.** "section 2.1 could be simplified, Make it concise please."

**Response:** The descriptions about reservoirs construction and the national nature reserve have been simplified in section 2.1.

**2.** "The m-k method used in this paper, which is the basis distinguishing method, please adding some distinguishing methods to make readers more convinced."

**Response:** Actually, some different methods have been used to verify and support the result of MK method in the previous manuscript. In section 4.1, the annual precipitation change between two periods was discussed by using MK method, and the result showed no significant trends, meanwhile, in section 4.3.2, the precipitation in summer and winter during three periods were compared (Table 7), and the relative changes were very slight, and in section 4.4, the LOR method was used to analyze whether the length of the collected data can eliminate the interference of climate on the IHA statistical results, and the result showed thar the data collected in this study basically satisfy the requirements for the analysis.

**3.** "The cascade reservoirs are built, the flow regime would be altered and not suited for spawning. Please clarify how and to what extent the former will impact the latter in your study area."

**Response**: After the construction of cascade reservoirs, the runoff averaging effect became more evident, which effected the transmission of fish spawning signals. The discussion on the impact of reservoir construction on fish reproduction in May and the restoration of ecological regulation have been added in Section 4.5 from line 379 to 387.

Special thanks to you for your good comments.

**Referee #5**

**1.** "In Table 6, the overall degree of alteration of group 1-4 is increased from 1998-2012 to 2013-2017, but the fifth group is decreased, please explain the reasons."

**Response:** In general, the impacts on the flow regime are further aggravated with the continuous construction of reservoirs, the averaging effect of runoff became more obvious, therefore, the group 5 (Rise Rate, Fall Rate and Number of Reversals) showed a decreasing trend in alteration degree during the period of 2013-2017. And The relevant discussion has been supplemented from line 296 to 306.

Special thanks to you for your good comments.

**The list of all relevant changes:**

**Line 1:** "analysing" was added

**Line 2:** "cascade" was corrected as "cascading"

**Line 10:** The method that we refer to is the conventional method of evaluating IHA parameters, and the relevant statement was added.

**Line 15:** The statement of "the continuous construction of reservoirs" was corrected as "the increase in the number of reservoirs".

**Line 16:** The explanation of different groups was added in the abstract.

**Line 17:** This statement of "a head drop and tail lift" was modified.

**Line 19-21:** The impact of ecological regulation has been added in the abstract.

**Line 26**: The estimated area was clearly stated.

**Line 28:** "was" was deleted.

**Line 28:** "from 1949" was added in the sentence, and 98795 is the number of dams built from the year when New China was founded to 2017.

**Line 29-30:** the values of the exponent was corrected.

**Line 31:** The correct citation has been provided, and the reference in the list has also been modified at line 461.

**Line 32, 34, 57,61, 63**: The references were listed chronologically.

**Line 77:** "with" was corrected as "of"

**Line 77:** This statement was modified

**Line 84:** "mm" was corrected as "mm/yr"

**Line 85**: "1/3" was corrected as "a third"

**Line 99**: "ecological" was corrected as "flow"

**Line 110:** This statement of "that larger values are better and that smaller values are better" was modified.

**Line 114:** "n" and "m" were defined, and the indicator and sample have been explained.

**Line 120:** This m is the same as the one above

**Line 129:** $z_i$ was defined at line 121.

**Line 132:** $n$ is the length of daily streamflow data, and it was defined at line 114.

**Line 136**: $t$ is the independent variable of the unit step function u(t).

**Line 147-170:** The application description of RAGA method was added.

**Line 174: "**Table 2" was corrected as "Table 3"

**Line 142**: s.t. is short for "subject to", and this means that the optimal projection direction must satisfy this condition.

**Line 186**: The repeated $i$ was replaced to $k$ and the repeated $n$ was replaced to $y$.

**Line 202-212**: The relevant description of the Mann–Kendall test method was added.

**Line 240:** "Table 3" was corrected as "Table 4".

**Line 246**: "formula" was corrected as "Equation".

**Line 262:** "of" was added.

**Line 263**: "when" was added and "is" was added.

**Line 296-308:** The relevant discussion based on the previous literature was supplemented.

**Line 325**:The statement of precipitation anomalies was supplemented.

**Line 379-387:** The discussion on the impact of reservoir construction on fish reproduction in May and the restoration of ecological regulation was added.

**Line 485:** A reference was added (quoted at line 212 of the manuscript).

**Line 526:** A reference was added (quoted at line 299 of the manuscript).

[revised manuscript text omitted]

**figure captions**

Fig 1. Generalized map of study region.

Fig 2. Precipitation changes in the Jinsha River Basin:(a) Trend of annual Precipitation at 28 stations between 1966 and 2017(Upward (downward) triangles indicate positive (negative) trends from MK test. The size of the triangles depicts the significance levels 10% (small), 5% (medium), and 1% (large). Black dots show stations with no trends or trends below 10% significance level. Value in the legend is the standardized statistics Z value).

Fig 3. Value of weights of 32 IHA parameters in the (a) Panzhihua Station, (b) Huatan Station, and (c) Pingshan Station. The red line is equal weight line (1/32=0.03125).

Fig 4. Project values in the (a) Panzhihua Station, (b) Huatan Station, and (c) Pingshan Station.

Fig 5. Correlation coefficients among the IHA statistics for the observed data sets in the (a) Panzhihua Station, (b) Huatan Station, and (c) Pingshan Station. (* means outlier.)

Fig. 6 Annual flow duration curves in 2004 and 2016 in the Panzhihua Station.

Fig. 7 Correlation between runoff and precipitation for the periods of pre– and post–impact in the Panzhihua Station.

Fig 8. Hydrograph for daily average inflow, outflow and reservoir water level in the (a) Xiluodu and (b) Xiangjiaba Reservoirs.

---

## Editor Decision (ED1)

[revised manuscript text omitted]
 1999-2012 Alteration Degree | Pingshan Station 2013-2017 Alteration Degree | Parameter | Pingshan Station 1999-2012 Alteration Degree | Pingshan Station 2013-2017 Alteration Degree |
|---|---|---|---|---|---|
| 1 | 71% | 100% | 17 | 86% | 100% |
| 2 | 86% | 60% | 18 | 0% | 60% |
| 3 | 86% | 100% | 19 | 0% | 60% |
| 4 | 43% | 100% | 20 | 14% | 60% |
| 5 | 14% | 100% | 21 | 43% | 60% |
| 6 | 14% | 20% | 22 | 0% | 60% |
| 7 | 43% | 60% | 23 | 14% | 100% |
| 8 | 0% | 20% | 24 | 57% | 60% |
| 9 | 14% | 20% | 25 | 14% | 60% |
| 10 | 57% | 60% | 26 | 29% | 100% |
| 11 | 0% | 60% | 27 | 86% | 100% |
| 12 | 0% | 60% | 28 | 57% | 20% |
| 13 | 43% | 100% | 29 | 14% | 60% |
| 14 | 57% | 100% | 30 | 57% | 20% |
| 15 | 86% | 100% | 31 | 100% | 100% |
| 16 | 100% | 100% | 32 | 100% | 100% |

[Figure]

[Figure]

Table 6. Overall degree of alteration of five groups of IHA parameters

[revised manuscript text omitted]